# The Potential of Utilizing Air Temperature Datasets from Non-Professional Meteorological Stations in Brno and Surrounding Area

**DOI:** 10.3390/s19194172

**Published:** 2019-09-26

**Authors:** Karel Dejmal, Petr Kolar, Josef Novotny, Alena Roubalova

**Affiliations:** Department of Military Geography and Meteorology, University of Defence, Kounicova 65, Brno 66210, Czech Republic; karel.dejmal@unob.cz (K.D.); petr.kolar2@unob.cz (P.K.); josef.novotny@unob.cz (J.N.)

**Keywords:** air temperature, amateur meteorological stations, seasonal variability, cluster analysis

## Abstract

An increasing number of individuals and institutions own or operate meteorological stations, but the resulting data are not yet commonly used in the Czech Republic. One of the main difficulties is the heterogeneity of measuring systems that puts in question the quality of outcoming data. Only after a thorough quality control of recorded data is it possible to proceed with for example a specific survey of variability of a chosen meteorological parameter in an urban or suburban region. The most commonly researched element in the given environment is air temperature. In the first phase, this paper focuses on the quality of data provided by amateur and institutional stations. The following analyses consequently work with already amended time series. Due to the nature of analyzed data and their potential use in the future it is opportune to assess the appropriateness of chronological and possibly spatial interpolation of missing values. The evaluation of seasonal variability of air temperature in the scale of Brno city and surrounding area in 2015–2017 demonstrates, that the enrichment of network of standard (professional) stations with new stations may significantly refine or even revise the current state of knowledge, for example in the case of urban heat island phenomena. A cluster analysis was applied in order to assess the impact of localization circumstances (station environment, exposition, etc.) as well as typological classification of the set of meteorological stations.

## 1. Introduction

Data from meteorological stations serve for evaluation of climatic situation, but also to give information about current weather and to predict its future course. In order to ensure maximum precision of the meteorological analysis, it is desirable to use a homogenous network of stations that is at the same time as dense as possible. Czech Hydrometeorological Institute (CHMI) operates over 800 meteorological and climatological stations [1] that should meet the standards established by World Meteorological Organization (WMO) [2].

In the Czech Republic there are numerous meteorological stations operated on amateur or semi-professional level by either individuals or institutions (expert offices of universities, city organizations, etc.), that are interested to collect and make available their own station data. Weather information could be also obtained from Amateur meteorological society (AMS) [3] that is in a long-term cooperation with CHMI, or directly from the owners of amateur meteorological stations. A significant increase of number of stations, namely notable in the proximity of population centers, can be attributed to the recent development and financial affordability of mobile applications. Their mapping services allow a real time visualization of meteorological information for a specific location in global scale [4]. Use of such kind of data represents a specific form of crowdsourcing [5], whose potential is, on the other hand, limited by a variety of factors. Amateur stations often do not conform to the WMO standard and are typically differing in measuring conditions as well as quality of measuring instruments.

An important stage in data processing is the control of their quality, consisting among others from definition of missing data periods, identification of errors in measurements and records. This phase is in longer-time series followed by series harmonization, which could be carried out through diverse statistical tests and instruments, described in several professional publications [6,7]. While aiming to procure complete records, it is necessary to supplement those values that are either missing, or cancelled due to detected errors.

The wide scope of climatological analyses gave base for a plurality of scientific studies that not only deal with statistical evaluation of air temperature, but also describe different methods and drawbacks of data processing and point to the issue of non-standard meteorological stations [8]. Amateur meteorological stations have already been used for some studies abroad e.g., estimation of existence of heat island inside a city in the Netherlands [9], or in the USA, where they provided a more complex information of the current weather conditions for tourists via internet [10].

The most influencing elements for air temperature measurement are, among others, sensor type and its placement (height, exposition, open/closed space), color, type and form of radiation cover, type of surface in the proximity of the sensor (natural/artificial) and the surroundings of the station in general (obstacles, surface diversity). A considerable effect, mainly in the summer season, occurs due to variability of solar radiation, but there are methods that ensure elimination of resulting uncertainties [11,12]. As evidenced in recent years, there are alternative methods of air temperature data gathering on the rise, for example through smartphone battery [13].

Besides meteorology and climatology, the air temperature records are used in other domains, such as transport and agriculture or integrated rescue system. Many researches nowadays focus on the topical phenomena of heat island formation inside urban areas [14,15,16,17]. Hand in hand with population growth, population density in cities and world urbanization rises the impact of urban environment on air temperature and other meteorological elements [18]. A dominant majority of publications, that for different objectives examine regional variability of air temperature, implements preferably data from professional meteorological stations. Many other studies work with satellite data that allow refinement of the temporal and spatial variability of surface temperature through infrared images [19,20,21]. Likewise, the preceding studies linked to Brno city and its close surroundings, mostly drew data from the network of standard meteorological stations [22], or, if appropriate, from ad hoc stations and line measurements from sensors located on vehicles, as well as remote sensing [23].

This study follows the previous study of usability of amateur meteorological stations in Brno and close surroundings [24] that examined a heterogeneous set of stations with emphasis on quality assessment of obtained data. This work aims to appraise the usability of amateur stations as well, which might in ideal cases supplement the network of professional stations and thus contribute to a spatial refinement of status and course of weather, help characterize the microclimate of specific localities or analyze temperature extremes. The first objective was to test the possibility of short-term outages supplementation in air temperature records, where several methods of temporal and spatial interpolation were compared. Another goal was to evaluate the seasonal spatial variability of air temperature in the area of interest and subsequent clustering of individual stations using multivariate statistical methods. In the Czech Republic this is a unique attempt to evaluate the potential and possible pitfalls of this kind of meteorological data.

## 2. Data and Methods

For air temperature variability analysis in Brno and surrounding area in the period 2015–2017, 14 meteorological stations were employed, applying maximum air distance of 25 km from the city center as a criterion of selection. The above-mentioned area is located in the Central European region, and in closer detail in the south-eastern part of the Czech Republic. Spatial distribution of the stations is demonstrated in Figure 1. It illustrates a non-homogenous group of stations within which amateur stations (operated by individuals) could be spotted, as well as stations operated by institutions of university or other character or the Czech Hydrometeorological Institute in professional mode.

The long-term average annual air temperature in this area ranges between 7.5–10 °C (monthly minimum −3–0 °C in January, maximum 17–20 °C in June) [25]. Changes in average annual air temperature has been showing an increasing trend of 0.3 °C/ 10 years since 1961 for the Brno-Tuřany professional station [23], that fully corresponds to other Central European stations [26]. The average annual sum of global solar radiation is 3800–4000 MJ·m^−2^ (monthly minimum 60–70 MJ·m^−2^ in December, maximum 560–600 MJ·m^−2^ in June), the average annual rainfall is 400–600 mm (monthly minimum 20–40 mm per month in the period from December to February, maximum 60–80 mm per month in June and July) and the reference evapotranspiration is between 700–800 mm per year (monthly maximum 120–140 mm per month in July) [25]. By the reference evapotranspiration is meant the evapotranspiration value of a hypothetical crop that closely matches the standard grass cover, with a uniform height 12 cm, full canopy closure, and optimum moisture conditions all year-round. According to Köppen climate classification [27], most of the analyzed area currently belongs to the climate subtype of deciduous forests of temperate zone Cfb, in the part of the territory with altitude exceeding 500 m the sub-type of boreal climate is Dfb [25]. With respect to the expected climate dynamics, most of the territory is likely to fall under the subtypes of the humid subtropical Cfa climate and the cold semiarid BSk climate by the end of the 21st century [28].

In total, there are 5 main reference groups and 8 soil types according to WRB classification [29] on the territory. Chernozems and Cambisols cover the largest part of the territory (together over 50%). While Chernozems are found in the lowland areas to the south of Brno city, the Cambisols are linked to more northern undulating areas. Luvisols spread on roughly 20% of the area (to the north of Brno) and the rest of the land is covered by Leptosols (northeast from Brno) and a stripe of Fluvisols [30] along Svratka river to the south of Brno. In the center of the area is Brno city (approximately 400,000 inhabitants) that is in the sense of land use/land cover (LU/LC) categorization made up of continuous and discontinuous urban fabric areas. Zones on the north from the city are predominated by broad-leaved and mixed forests areas while southern parts are characteristic by “annual crops with permanent crops” category. The other surface types (water bodies, vineyards, airports, mineral extraction sites, etc.) take up insignificant portions of the land [31]. In the three-year horizon between 2015 and 2017 only very minor changes in LU/LC occurred in the region, mostly in relation to the suburbanization and industrialization processes around Brno. Neither of the changes took place 500 m or closer to any of the involved meteorological stations.

A more detailed overview of the stations is shown in Table 1. Stations recorded data in various time steps that were converted into 1 h step for consequent analysis. A seasonal time step according to established standards (winter: December-February, spring: March-May, summer: June-August, autumn: September-November) was chosen for more detailed evaluation of temperature data variability. While amateur stations are operated on voluntary base and any relevant maintenance (sensor calibration, adaptation of station surroundings) depends on the decision of an individual (station owner), the institutional stations provide data for specific purposes and should therefore comply with higher qualitative standards. Lastly, professional stations observe general standards [2], the operating personnel is adequately instructed through standard operating procedures [32] and resulting meteorological data are subject to a fee.

Quality control of the data was the first step of their processing. The air temperature course for every station was drawn in a monthly step. Consequently, this was supplemented by the air temperature course of the reference station BTp, which had been selected from the group of professional stations. If accessible, interval minima and maxima of air temperature were then visualized. This subjective method was applied to identify the most frequent types of errors and most problematic stations. The resulting detection of suspicious values was based on the following objective parameters: (1) Air temperature values fall outside the interval −50–50 °C; (2) interpolated value (bilinear interpolation) from the two neighboring values differs from the real value by more than 15 °C (only counted if the time difference of the near observation times does not exceed 31 min); (3) occurrence of constant values—if there are at least 5 consecutive identical values within a 2 h time window; (4) if the values of the controlled series differ from the reference series by more than 10 °C (only counted if the available time in the reference series does not exceed 20 min from the time of observation); (5) air temperature is lower that the dew point temperature (in case the dew point temperature is available); (6) air temperature is lower than minimum or higher than maximum air temperature (if those are available for the interval).

Group of suspicious values was then finally evaluated and verifiable errors were removed from the series. The algorithm did not succeed in detecting all data, which were removed in the end. On the contrary, it pointed to many values that were probably correct. Problems occurred for example due to the unique exposition or various obstacles nearby the stations). Therefore, the data were not representative for a larger vicinity of the station, but only for the given atypical site. This led us to carry out a thorough field exploration of all stations, metadata collection, photographic documentation and geodetic measurement. A confrontation with valid meteorological standards [2,32] brought about following findings: occurrence of obstacles (such as buildings, trees) modifying the radiation and wind conditions within a distance of 4 times the height of the obstacle from the station (Figure 2);the height of the temperature sensor over the ground surface;type of the radiative shield (standard—white metal or plastic shield, parameter had not been found);sensor location (such as open space, wall of a building—the sensor is positioned directly on the wall or at a maximum distance 50 cm);type of ground surface (P—paved, G—grassy, M—mixed) in a distance < 5 m;spatial exposure (no obstacles occur in the given direction).

The observed parameters are illustrated in Table 2. No technical parameters were considered (sensor manufacturer, measurement accuracy, calibration frequency, etc.).

Frequency of failures is another indicator of data usage. It has been evaluated after the interpolation to whole hours was finished (see below), since some stations either recorded the data in such fixed observation times that never or rarely corresponded with whole hours, or were recorded in an irregular time step.

One of the possible ways to supplement the missing data is to apply a temporal interpolation from the existing data of the given series. The accuracy of interpolation depends not only on the interpolation method, but also on the period to which data are interpolated. For our case the selected interpolation method is bilinear interpolation from neighboring values [33]. A main disadvantage of such method is its incapability to interpolate extremes. On the other hand, the fact that it does not generate unrealistic values is a clear benefit. Other methods were considered to compare the accuracy—to take the last available value (value of the beginning of the given interval) and the last one was calculation of the cubic splines [34]. In order to estimate the expected accuracy of interpolation, a test of accuracy of above stated methods has been executed on stations with 1-min data (Hos, Rou and Zby). The tests worked with different time intervals in between the given data up until a duration time of one hour. Interpolation was realized for all possible cases inside this interval. E.g. for a 5-min interval we tested the interpolation accuracy of 1, 2, 3 and 4 min after the beginning of interval. All possible intervals of this length that could have been derived from the data, which offered a comparison between interpolated and real value, had been considered. The standardly applied characteristics were counted for the accuracy appraisal: RMSE (root mean square error) and values of 95% and 99% quantile of error (absolute value of difference). Consequently, a dependence of those characteristics on the time of day has been surveyed.

As there were several time outages longer than one hour that could not have been filled by temporal interpolation, the possible usage of data from neighboring stations was elaborated. An optimal number of most suitable stations had to be chosen for this calculation. A set of most similar stations was defined for each station. Appropriateness of stations for the calculation was assessed through: (1) Coefficient of correlation, (2) standard deviation of value differences on the stations, (3) Euclidean distances from the cluster analysis and (4) geographic distance. Five stations were always selected on the base of those criteria [7]. We determined five air temperature estimates by adding the average difference between the calculated station and the given station for each of the five stations. The resulting air temperature estimate was done and presented on the average of these five estimates (no weights were considered). Similarly, other combinations of selection from these stations were tested—from four stations, omitting one of them, three and two most similar stations, and each of five stations individually.

This procedure has been applied on all data without distinction of time, but also individually for every hour of the day. The role of selection of stations individually for every hour in contrast to the selection based on all data and their use for whole day has also been subjected to a test. After the appraisal of interpolation accuracy (RMSE) for observation times with available observations, the best method as well as the optimal set of stations for every station was selected and missing values were calculated. In case the best combinations of stations could not have been used due to failures on one of them, the next best available combination was applied respecting the order of successfulness.

For each of the 14 stations the completeness of given time series including data filled through interpolation was assessed. Consequently, a file containing the intersection of data from all 14 stations was generated. The ratio of missing values from the diurnal course was assessed firstly, and in the end the evaluation assessed data availability for individual seasons. In order to illustrate temperature variability of stations, 6 of them were typologically singled out and diurnal course of air temperature was shown with distinction of individual seasons. The set of stations in question contained purely urban stations (BZa, Met), but also suburban (Tro) and rural (TiH, Zab) stations and even a station specific for its valley location (Hos). Pearson’s correlation coefficient has been calculated for air temperature time series for the whole set of 14 meteorological stations values. Statistical significance has been evaluated for every combination of pairs of stations on the confidence level p = 0.05.

A cluster analysis was carried out on the common data of all stations in order to express the rate of relative similarity of air temperature time series from individual stations. The analysis figures among explorative multidimensional statistical techniques that are often used in climatology to define the climatic zones e.g., [35]. In case of smaller territorial range, the results of cluster analysis may serve for clustering of stations with similar standpoint or exposition conditions [36]. The aim is to reduce the total number of stations into several clusters, where the stations are as similar as possible and stations outside of a given cluster are as different as possible. The cluster analysis offers a larger number of possible ways to arrange the stations into clusters. Two methods were selected based on the recommendations provided by climatological studies [37]—non-hierarchical clustering via k-means and hierarchical clustering using Ward method. A quantified rate of difference is Euclidean distance that can be calculated between either individual clusters or among all stations. To ensure the representativeness of results from given methods only those hourly observation times were selected in the period 2015–2017, which disposed of available data from all stations. Software STATISTICA 12.0 was employed for data processing.

## 3. Results

At least one incorrect value was removed on 4 from 14 stations. This concerned stations Arb (0.38% values removed), Met (< 0.01%), VDB (0.66%) and Zab (< 0.01%). The most frequent cause of removal was the occurrence of constant values or continuous significantly varying data ended with an outage or abrupt change. In other cases, it was mostly individual notably differing values. For some stations, in the case of measurement failure (or record failure), a defined value falling outside of the range of real values appears. Such value, e.g., in Arb station, was not identified until this control, which means it is not a result of faulty measurement but measurement failure.

As far as the estimated accuracy of temporal interpolation, the results for individual stations were comparable (Figure 3). All the stations encountered worst quality results around midday and then the characteristics gradually decline. Similarly, winter season displayed better quality of interpolation in comparison with summer and also spring (Figure 4a). Maximum values of RMSE for interpolation increased within selected length of interval. Similarly, depending on the length of interval, maximum quantile (Q95, Q99) values rose as well. The worst results inside one time interval appeared close to the middle of interval (Figure 4b). The more the values approached to one of the limit values, the more rose the accuracy. Bilinear interpolation returned the best results in general. Moreover, the same conclusion could be stated considering the cases of only few minutes after last available recorded value when comparing to the second method of last available value. Cubic splines offered slightly worse interpolation than bilinear interpolation for most of the stations, although there were only minor mutual differences. If compared the dependence of characteristics (RMSE, Q95, and Q99) on the time of day, there were no significant differences among the stations.

Temporal interpolation was applied to supplement missing values on full hours for all stations on such event, where the window between the available values around this hour was maximum 60 min and one of those values had been recorded maximum 10 min from the given observation time. On the base of above described tests on 3 stations the resulting estimated value of RMSE in the worst case was 0.2–0.3 °C, Q95 around 0.5 °C and Q99 0.8–1.0 °C. No value was filled with this technique on 11 stations from 14.

Spatial interpolation helped to supplement all data. Calculation via mean difference of air temperatures without distinction to the time of a day turned out to be substantially worse than calculation with a link to every hour separately. In 10 cases the best criterion for the selection of stations proved to be the standard deviation (Table 3). Coefficient of correlation was more appropriate in 2 cases and geographical distance in 2 cases. The difference between using the same relationship for a whole day and individual relationship for every hour was not distinct but the best methods were always individual selection. The optimal number of neighboring stations varied among the stations. In the case of Dav there was only one selected station. Selection of two stations resulted five times, of three and four stations thrice and of five stations twice. RMSE would decline with the rising number of selected stations during elimination of cases of neighboring stations (Met, Dav, Arb). Mean RMSE value amounted to 0.8 °C and ranges from 0.3 °C (Met) to 1.8 °C (Mok), mean value of Q95 is 1.6 °C (0.5–2.4 °C) and Q99 2.8 °C (1.1–6.2 °C). Results for the examined stations are therefore considerably worse than would correspond to the range of temporal interpolation. On the base of obtained expected interpolation errors the missing data in time series were filled only through bilinear interpolation and other tested approaches were not applied.

Stations BTp, BZa, TiH, Tro, Hos, Rou and thanks to the temporal interpolation Zby as well, show all data from the period 2015–2017 (26 304 hourly values). Neither of the stations exceeded the 20% threshold of missing values, the closest being Mok (16.3% of missing values), Siv (10.2%) and Dav (7.4%). In terms of diurnal course, the data availability for individual hours of the day varied only insignificantly and in the 24-h slots ranged from 70.1% (09:00 CET) to 72.7% (02:00 CET). The results however differed more significantly by season. The highest ratio of serviceable hourly data appears in the summer (81.8%), then spring (73.6%) and autumn (70.3%) followed by winter season data (60.8%).

Diurnal air temperature courses (Figure 5) contain, for all seasons and all meteorological stations, one afternoon maximum and one morning minimum. Hand in hand with the time of sunrise the temperature maxima in the winter occurred 1–2 h later than in the summer. Apart from Siv station the daily minima in the winter occurred at 7:00 CET. In the summer the recorded variance was more prominent, the average temperature minima occurred at 4:00 on 5 stations, at 5:00 on 8 stations and at 6:00 CET on 1 station. Summer daily temperature maxima on the other hand shifted to later afternoon. While in the winter the daily maxima on almost 65% of stations occurred at 14:00 CET, the maxima in the summer occurred at 15:00 on 80% of stations. In both seasons the maximum at Siv station was recorded one hour earlier and on stations Mok and VDB one hour later. Daily temperature amplitudes ranged from 3.3 °C (TiH) to 4.4 °C (Zab) in the winter and from 10.1 °C (TiH) to 12.9 °C (Hos) in the summer. The urban stations (such as BZa, Met) in comparison with other stations registered systematically higher air temperatures at night regardless of the season. On the contrary, a significantly lower air temperatures at night and in the morning were detected on station Hos. In the summer, the temperature difference from second coldest station between 20:00 and 06:00 CET even consistently surpassed 2 °C. Calculation of Pearson’s correlation coefficient for all pairs of air temperature time series demonstrated presence of statistically significant relations on a selected level of confidence. Values ranged from r = 0.963 for the pair Hos–Mok to r = 0.999 for the pair Day–Met.

The results of cluster analysis prove that geographical distance of stations is not a crucial factor of shared variability in the air temperature data. A more notable impact could be attributed to the sensor placement (surface nearby the stations, city/landscape) or microclimatic conditions. The group of 14 stations can be seasonally divided into several clusters through the application of Ward method that additionally brings also the genesis of clustering. Winter season (Figure 6a) brings an apparent mutual similarity of urban stations (BZa, Dav, Met, VDB) on one side and rural stations (Tro, Zab, Rou, Siv) on the other. The third cluster is formed by the rest of the stations except Hos that demonstrates the most notable difference from other stations throughout all seasons. Temperature data for spring season (Figure 6c) evidence a shift of BTp station to the urban cluster. Also, the mutual variance within rural stations cluster increases. A similar pattern is noticeable for summer season as well, aside from the fact that urban stations cluster is divided into two subgroups: the first subgroup comprises stations from the eastern part of the city (BTp, Met, Dav) and second one from the western part (BZa, VDB) of Brno. For spring, summer and autumn there is a separate cluster formed by stations Arb, Zby and TiH, whose common denominator is neither geographical distance nor typology of environment (city x countryside), but a placement of the station on the upper section of a slope.

## 4. Discussion and Conclusions

Data from amateur and institutional stations suggest a remarkable potential of application in a wide variety of domains [5]. Studies comparing measurement results from professional meteorological stations and measurement sets generally accessible for amateur meteorologists did not identify any significant differences for the majority of parameters, including air temperature [38]. In spite of that, it is always essential to subject all data to an individual quality assurance and quality control before proceeding to an analysis. Bearing in mind the specificity of a given task, it is usually more opportune to rather seek a declaration of quality of used stations than use maximum possible number of available stations [39]. The previous phase of quality assessment of data from a group of amateur and institutional stations in Brno and surrounding area [24] identified and removed those stations, whose obtained air temperature data, are generally subject to the most common type of errors, such as constant or physically inconsistent values. Another frequent challenge of amateur meteorological stations are biased values due to an inadequate radiation cover or limited sensor ventilation [8]. Such stations were also eliminated from analyses of air temperature seasonal variability.

Despite the above mentioned procedures, it can be expected that the analyzed air temperature time series of the resulting set of meteorological stations are burdened with uncertainties. The quality control process of the measured data detected the most problematic values, but could not filter out any possible errors. The most problematic aspect of amateur meteorological data is the absence of regular sensor calibration [8], which is mainly due to ignorance of its necessity or the financial and logistical complexity of the calibration process. The quantification of uncertainties associated with uncalibrated sensors would require long-term parallel measurements at each site. The representativeness of the station location can be considered as another source of uncertainty because it is questionable what spatial scale a given station represents [40]. Based on the results of the analyses, an example of a station, whose location represents specific microclimatic conditions can be given. The spatial range of Hos station is limited to inverse valleys of karst relief in the northeastern part of the area of interest. The results of the cluster analysis confirm that it is the station that differs most from the set of remaining stations. Another source of uncertainty is the diversity of types of technical equipment of meteorological stations, sensors and radiation shields. Nevertheless, in special-purpose measurement campaigns various types are used [41].

Temporal interpolation is a more appropriate method of adding missing values than spatial interpolation even for a set of nearest stations. It would be preferable to verify the suitability of both methods even in longer time intervals than one hour. The accuracy of time interpolation can be expected to deteriorate with increasing time intervals. At the same time, it is possible to test other temporal interpolation approaches (e.g., regression models). The results of time interpolation are related to uncertainties arising from the assumption that the daily air temperature curve is unchanged. It would be better to parameterize the shape of the diurnal course curve [42], especially for longer time periods of values added through calculation (30 and 60 min, respectively). It was demonstrated that in the spatial interpolation from the given stations the key role is played by the degree of stations’ similarity which is determined by the size of the standard deviation of the difference of values or the correlation dependence, and not geographical or Euclidean distance between stations. A different spatial interpolation provides better results for each daytime separately. In most cases, it turned out to be better to limit the selection for less than five stations, which is a generally recommended number when working with professional station data [6].

The results of field survey confirm that the stations are typologically very diverse. The differences are induced by their different parameters resumed in Table 2. Some of those parameters in the given urban setting influence the meteorological data more significantly than basic physical-geographical parameters (altitude, longitude, topography). This mostly concerns type of surface and exposition. That is why it is very difficult to interpret data in sum and in context of other works to discuss, e.g., the issue of the urban heat island [16]. Nevertheless, the analyzed data imply that urban stations register higher temperatures throughout the year than other stations. The mutual differences are brought about by higher temperature minima rather than maxima (Figure 7), which is linked to the urban structures, radiating heat all around at night but on the contrary creating shadows during the day. Thanks to the long-term quality measuring on BTp station, this is the station that often provides referential data for the region [23]. Results of the analysis however showed, that for the purpose of urban heat island quantification this is not the ideal station. The character of its surroundings (impermeable airport tracks) resembles that of urban stations, which are influenced by night radiation from artificial surfaces (Figure 6). This influence gains on importance in case of radiative type of weather during spring and summer period. The most apt referential station in light of the former reasoning is therefore Zab, whose location ensures a substantial minimization of anthropogenic influences.

A short period of time prevents the use of data for climatological purposes. Nevertheless, it is possible to present the basic statistical characteristics of the period 2015–2017 on the basis of complete time series, supplemented by temporal or spatial interpolation data (Figure 7). No significant discrepancies were found. Stations located at higher altitudes north of Brno (TiH, Hos) are cooler and the data are also affected by inverse valley location of the station Hos compared to other stations. As far as urban stations, VDB could be labelled the hottest station, which can be attributed, besides other factors, to its placement in a relatively open terrain very near Brno dam. Except for spring, the proximity of water surface causes local increase of air temperature at night.

Comparison of three nearby stations in Cerna Pole (Ard, Dav, Met) confirms that the physical distance or altitude is not the most important parameter within the analysed stations’ set. For a more detailed interpretation, it is necessary to analyze the data under specific meteorological situations (radiation × advection days, etc.) at different times of the day, which would be the aim of the following research issues.

## Figures and Tables

**Figure 1 sensors-19-04172-f001:**
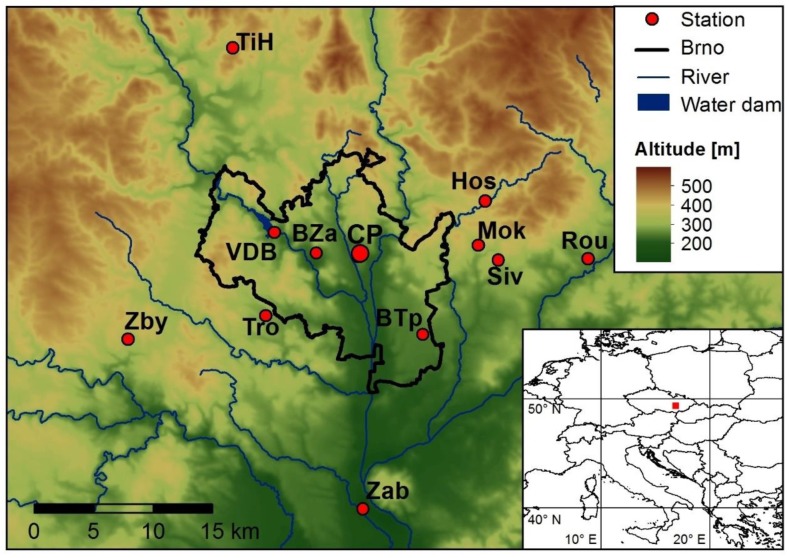
Spatial localization of employed meteorological stations in Brno and surrounding area; abbreviations of station names are explained in Table 1 (CP–locality Černá Pole, containing a total of three stations–Met, Dav, Arb).

**Figure 2 sensors-19-04172-f002:**
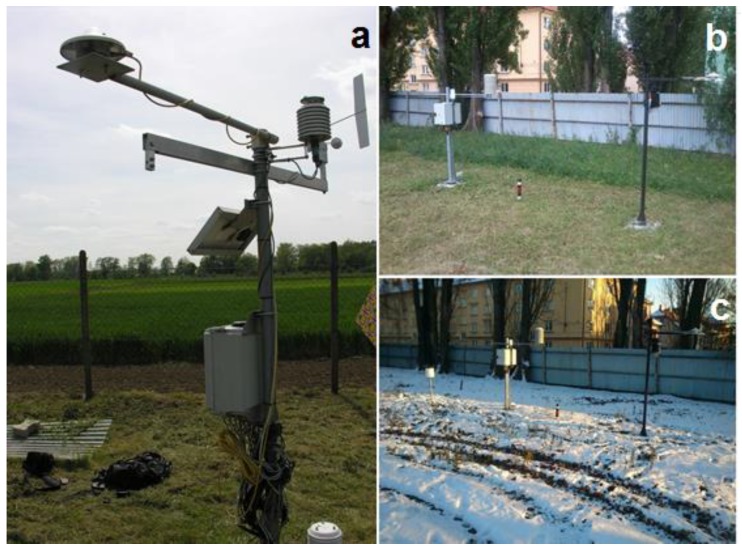
Diversity of the surroundings and occurrence of barriers on the example of stations Zab (**a**) and Met in the summer (**b**) and winter (**c**) period.

**Figure 3 sensors-19-04172-f003:**
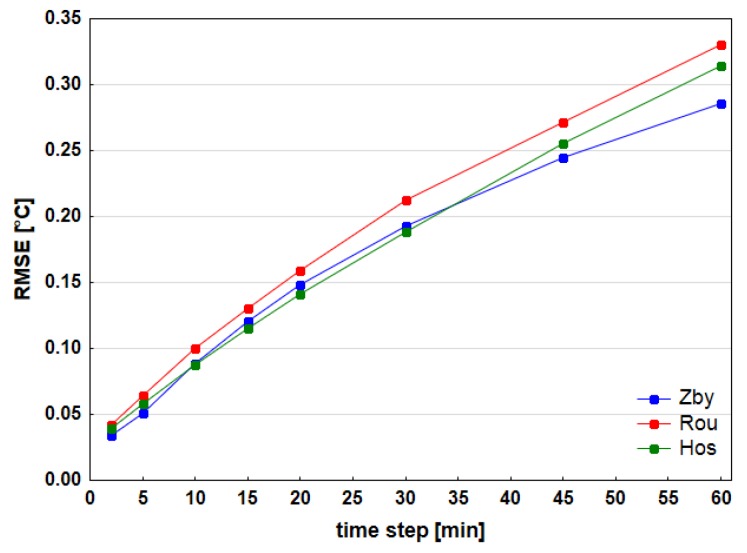
Comparison of accuracy of bilinear interpolation of 1-min air temperature data from 5-, 10-, 15-, 20-, 30-, 45- and 60-min time intervals on meteorological stations Zby, Rou and Hos.

**Figure 4 sensors-19-04172-f004:**
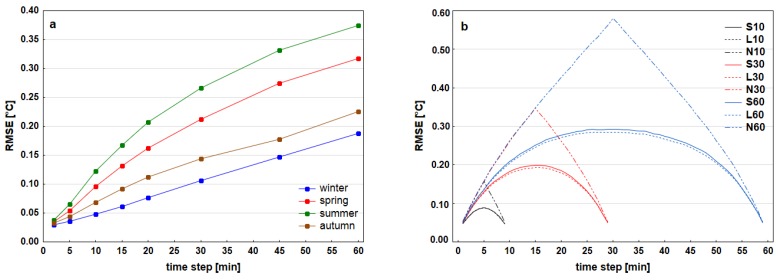
Seasonal accuracies of bilinear interpolation from 5-, 10-, 15-, 20-, 30-, 45- and 60-min time intervals (**a**) and comparison of accuracy of different time interpolation methods of 1-min air temperature data from 10-, 30- and 60-min time intervals on meteorological station Zby. (**b**): Bilinear interpolation (L10, L30, L60), last available value method (N10, N30, N60) and cubic splines interpolation (S10, S30, S60).

**Figure 5 sensors-19-04172-f005:**
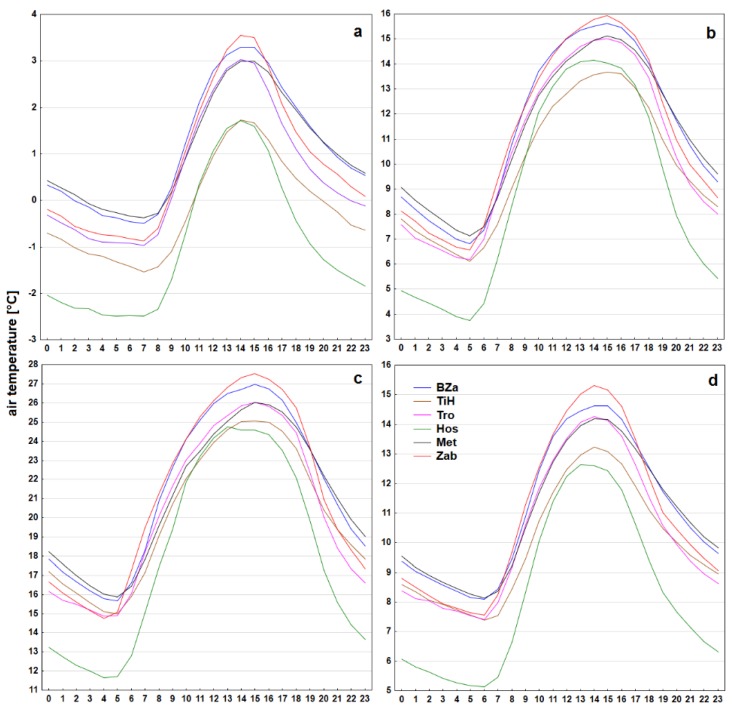
Seasonal (**a**—winter, **b**—spring, **c**—summer, **d**—autumn) daily temperature courses on stations BZa, TiH, Tro, Hos, Met and Zab for the period 2015–2017; the time axis corresponds to CET.

**Figure 6 sensors-19-04172-f006:**
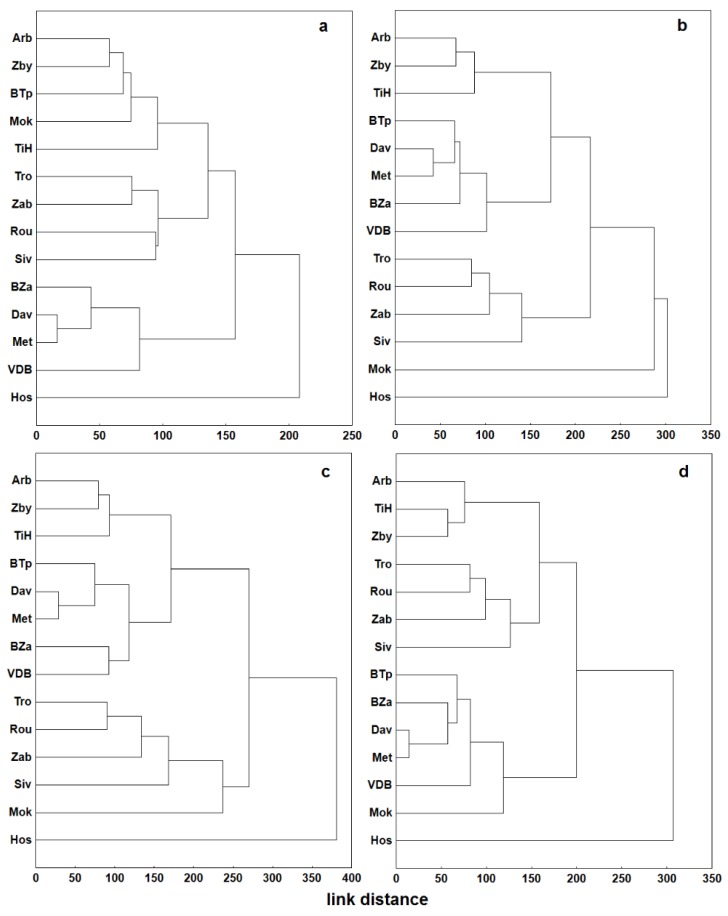
Seasonal (**a**—winter, **b**—spring, **c**—summer, **d**—autumn) cluster analysis dendrograms for the set of meteorological stations in the period 2015–2017 from common hourly data of air temperature.

**Figure 7 sensors-19-04172-f007:**
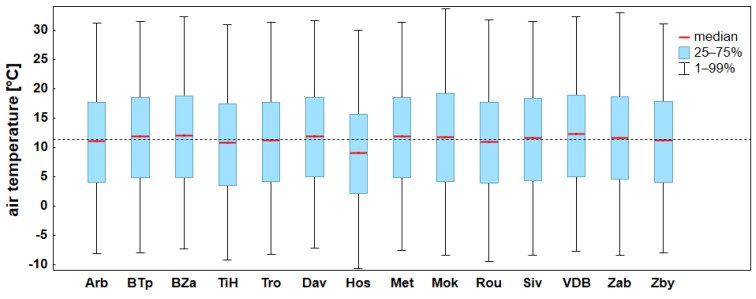
Basic statistical characteristics of air temperature for individual stations in the period 2015–2017 (the dashed line corresponds to the average air temperature of all the stations).

**Table 1 sensors-19-04172-t001:** List of used stations and their characteristics (station type: A—amateur, I—institutional, P—professional).

Station Name	Abbr.	Station Type	Altitude [m]	Measurement Interval [min]
ČP Arboretum	Arb	I	247	15
ČHMÚ Brno-Tuřany	BTp	P	241	30
ČHMÚ Brno-Žabovřesky	BZa	P	236	30
ČHMÚ Tišnov-Hájek	TiH	P	409	30
ČHMÚ Troubsko	Tro	P	278	30
ČP DAVIS	Dav	I	248	5
Hostěnice	Hos	A	351	1
ČP METEOS6	Met	I	248	10
Mokrá	Mok	I	325	60
Rousínov	Rou	A	244	1
Sivice	Siv	I	300	60
VD Brněnská	VDB	I	236	60
Žabčice	Zab	I	180	10
Zbýšov	Zby	A	345	1

**Table 2 sensors-19-04172-t002:** Summary of parameters influencing the recorded air temperature data according to [2] and [32].

Station	Obstacles	Sensor Height [m]	Radiative Shield	Placement	Surface	Exposition [°]
Arb	YES	2	standard	space	G	340–290
BTp	NO	2	standard	space	G	0–360
BZa	YES	2	standard	space	M	160–180; 260–90
TiH	YES	2	standard	space	G	0–270
Tro	YES	2	standard	space	M	280–230
Dav	YES	1.7	standard	space	M	330–30
Hos	YES	2	standard	space	G	260–80
Met	YES	2	standard	space	M	330–30
Mok	YES	2	standard	space	G	320–260
Rou	YES	2	standard	space	M	235–145
Siv	NO	2	standard	space	M	0–360
VDB	YES	1	?	space	G	0–180
Zab	NO	2	standard	space	G	0–360
Zby	YES	2.7	standard	wall	M	360–90

**Table 3 sensors-19-04172-t003:** Overview of best methods of spatial interpolation for individual stations (CC—correlation coefficient, D—geographical distance, SD—standard deviation).

Station	Method	Optimal Number of Neighboring Stations	RMSE [°C]	Q95 [°C]	Q99 [°C]
Arb	CC	3	0.4	0.9	1.4
BTp	CC	5	0.7	1.5	2.4
Bza	D	4	0.5	1.0	1.5
TiH	SD	2	0.9	1.8	2.6
Tro	SD	3	0.8	1.6	2.8
Dav	D	1	0.4	0.5	1.8
Hos	SD	2	1.2	2.4	3.5
Met	SD	2	0.3	0.6	1.1
Mok	SD	5	1.8	2.8	6.2
Rou	SD	4	0.9	1.8	2.8
Siv	SD	4	1.2	2.4	3.6
VDB	SD	2	0.9	1.9	3.0
Zab	SD	2	1.1	2.2	3.4
Zby	SD	3	0.7	1.4	2.2

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
