# Peer review of "The Potential of Utilizing Air Temperature Datasets from Non-Professional Meteorological Stations in Brno and Surrounding Area"

_sensors, 2019, doi:10.3390/s19194172_

Round 1
Reviewer 1 Report
This manuscript is dedicated to an important topic of air temperature assessment from the network of amateur/semi-professional meteorological stations. The research shows that this kind of stations has a sound potential for a number of climate- and weather-related applications. The authors made a laudable effort of explaining the observed temperature differences and pinpointing the experimental artefacts responsible for them.
I believe that the manuscript can be published in the journal given that the following questions are addressed. I put it to category “minor revision” believing that this will further improve the manuscript.
Main comments:
The temporal interpolation is performed as if the diurnal cycle were an abstract mathematical function whereas it follows a certain physically driven sequence of processes. Its shape (e.g. Aires et al., 2004) can be parameterized and the fitting can be done like in (Feofilov and Stubenrauch, 2019) or using any other suitable fitting technique. Moreover, one can even derive the parameterization from Fig. 2 of the manuscript where all curves demonstrate the same pattern. Cluster analysis is a valid approach, but in my opinion this section of the manuscript is overloaded and the information in Fig. 3 is difficult to read. I do not see an advantage of this methodology over the Pearson’s correlation coefficient (or r.m.s. of the difference) calculated for all pairs of stations and organized in a rectangular or triangular color matrix where the highest correlation coefficient is marked with red and the lowest one with blue – this plot is much easier to read and the cross-correlation of stations pops up by itself. I understand that the authors have developed and honed the methodology they describe, but it takes more time to grasp and the plots obtained with its help are less informative. Last but not least: the analysis has been performed, the data from all stations have been understood, but no recommendation on how to merge them to a single value and report to higher level network is given. For example, it is not clear whether we have enough information to build a long gapless sequence of air temperatures in Brno and surrounding area to establish a temperature trend. Before we do this, we have to understand, what we call an “air temperature in Brno”. It is clear that this value will come from a weighted sum of temperatures measured by individual stations, but how to mix them up in such a way which would not introduce a trend or bias given that the number of stations changes with time?Other comments:
Lines 113-130: the filtering is needed, and the approach seems to be logical, but how to ensure that the filtering does not introduce a trend or bias? In the current form, this paragraph looks qualitatively good, but the numbers and estimates are missing.
References
Aires, F., Prigent, C., and Rossow, W. B.: Temporal interpolation of global surface skin temperature diurnal cycle over land under clear and cloudy conditions, J. Geophys. Res., 109, D04313, https://doi. org/10.1019/2003JD003527, 2004.
Feofilov, A. G. and Stubenrauch, C. J.: Diurnal variation of high-level clouds from the synergy of AIRS and IASI space-borne infrared sounders, Atmos. Chem. Phys. Discuss., https://doi.org/10.5194/acp-2019-166, in review, 2019
Author Response
You can find the point-by-point response to comments of reviewer no.1 in the attachment.

Reviewer 2 Report
I found the paper has as interesting goal to be pursued and potential for its publication on the Sensors Journal, but not in the present form. The contribution of the paper is not clear with respect to previous works published in the literature. The reviewed literature need to be enlarged and improved.
The authors chose some options for the methodological approach but they do not provide the reader with enough background information, results and discussion for the results. The technical approach itself needs to be adequately addressed to show clearly original contribution. Clustering analysis procedures proposed are currently used in different applications.
The problem is that the use the procedures previously mentioned do not support the claim made by the authors in their summary with respect to adequately represent spatial-temporal air temperature. Complementarily, there is no use of any remotely sensed images to address evolution of land use and soil cover and surface temperature and its corresponding relationship with air temperature. Calibration and validation procedures are not clearly presented. Additional figures are necessary to better expose results.

Author Response
You can find the point-by-point response to comments of reviewer no.2 in the attachment.

Round 2
Reviewer 2 Report
Title: Seasonal Variability of Air Temperature on Various Meteorological Stations in Brno and Surrounding Area
New proposed title: The Potential of Utilizing Air Temperature Datasets from Non-professional Meteorological Stations in Brno and Surrounding Area
As requested, I have reviewed the second version of the above-titled paper for potential publication in the Sensors Journal. Please, see file attached.
